# RIOK2 Contributes to Cell Growth and Protein Synthesis in Human Oral Squamous Cell Carcinoma

**Yusuke Matsuzaki [1,2], Yutaka Naito [1,*], Nami Miura [1], Taisuke Mori [3], Yukio Watabe [4], Seiichi Yoshimoto [5], Takahiko Shibahara [2], Masayuki Takano [2] and Kazufumi Honda [1,6]**

1. Department of Bioregulation, Institute for Advanced Medical Sciences, Nippon Medical School, Tokyo 113-8602, Japan
2. Department of Oral and Maxillofacial Surgery, Tokyo Dental College, Tokyo 101-0061, Japan
3. Department of Diagnostic Pathology, National Cancer Center Hospital, Tokyo 104-0045, Japan
4. Department of Dentistry and Oral Surgery, Tokyo Metropolitan Tama Medical Center, Tokyo 183-8524, Japan
5. Department of Head and Neck Surgery, National Cancer Center Hospital, Tokyo 104-0045, Japan
6. Department of Bioregulation, Graduate School of Medicine, Nippon Medical School, Tokyo 113-8602, Japan
* Correspondence: y-naito@nms.ac.jp; Tel.: +81-3-5814-6913 (ext. 5588)

**Abstract:** Ribosomes are responsible for the protein synthesis that maintains cellular homeostasis and is required for the rapid cellular division of cancer cells. However, the role of ribosome biogenesis mediators in the malignant behavior of tongue squamous cell carcinoma (TSCC) is unknown. In this study, we found that the expression of RIOK2, a key enzyme involved in the maturation steps of the pre-40S ribosomal complex, was significantly associated with poorer overall survival in patients with TSCC. Further, multivariate analysis revealed that RIOK2 is an independent prognostic factor (hazard ratio, 3.53; 95% confidence interval, 1.19–10.91). Inhibition of RIOK2 expression by siRNA decreased cell growth and S6 ribosomal protein expression in oral squamous cell carcinoma cell lines. RIOK2 knockdown also led to a significant decrease in the protein synthesis in cancer cells. RIOK2 has potential application as a novel therapeutic target for TSCC treatment.

**Keywords:** RIOK2; biomarker; tongue squamous cell carcinoma; ribosome; protein synthesis

## 1. Introduction

Head and neck squamous cell carcinoma (HNSCC), which arises from the oral cavity, pharynx, and larynx, is the sixth most commonly diagnosed cancer in the world [1]. Tongue squamous cell carcinoma (TSCC) is among the most prevalent of the HNSCCs that arise from the mucosal epithelium of the tongue, and its incidence rate has been increasing globally [2]. Indeed, TSCC accounts for approximately 40% of all oral cancers [2]. Despite improvements in therapeutic strategies and diagnostic technologies, the 5-year survival rate of patients with TSCC remains poor [3]. Therefore, it is necessary to explore potential therapeutic molecular targets that will improve survival in TSCC patients. Moreover, surgery for TSCC can cause aesthetic and functional impairments that decrease postoperative quality of life (QOL). Biomarkers that can accurately predict outcomes in patients with TSCC can support better decision-making with regard to therapeutic strategies and follow-up options, leading to improved QOL in these patients.

Ribosomes are found in all living cells where they are responsible for the protein synthesis that maintains cellular homeostasis. There is increasing evidence that cancer cells increase ribosome biogenesis in order to meet the demand for the increase in protein synthesis that is required for rapid cellular division [4]. Hence, several reports have investigated the potential application of inhibition of ribosome biogenesis and function as a treatment strategy for cancer. For example, several anti-cancer drugs, including platinum-based DNA-damaging agents, interfere with ribosome biogenesis [5,6]. Several RNA polymerase I inhibitors that disrupt ribosomal DNA transcription have been developed

and are well characterized [4]. However, the mechanisms by which the mediators of ribosome biogenesis contribute to cancer progression remain unclear.

Rio-kinase 2 (RIOK2) is an atypical serine-threonine kinase that participates in the nuclear export and maturation steps of the pre-40S ribosomal complex to facilitate cytoplasmic translation [7–10]. RIOK2 binds the pre-40S subunit and promotes its nuclear export by interaction with the CRM1 chaperone protein [11]. Depletion or loss of catalytic activity of RIOK2 prevents maturation of the pre-40S subunit [11]. Recent studies have reported RIOK2 overexpression in non-small-cell lung cancer (NSCLC) and glioma [12–14]. Moreover, RIOK2 inhibition has been shown to decrease cell proliferation and induce cell-cycle arrest and apoptosis in cancer [13–15]. Although several reports have demonstrated the consequences of RIOK2 inhibition in cancer progression, it remains unclear whether RIOK2 expression contributes to ribosomal function and progression in TSCC. We recently developed a novel screening system for antibody-based proteomics, termed automated quantitative virtual immunofluorescence pathology (AQVIP), which can identify potential biomarkers involved in the patient's outcome based on the expression level of the targeted molecule. The system employs a computational system that automatically analyzes pathological images and clinical information [16].

In this study, we explored molecular candidates for TSCC treatment using AQVIP to investigate whether RIOK2 expression is associated with the progression of TSCC. In addition, we investigated the biological behavior of RIOK2 for acquisition of the malignant phenotype in oral squamous cell carcinoma cell lines.

## 2. Materials and Methods

### 2.1. Patients and Tissue Samples

We examined the formalin-fixed paraffin-embedded (FFPE) surgical tissue specimens of 40 patients with TSCC who underwent glossectomy alone with curative intent for stage-I/-II TSCC at the National Cancer Center (NCC) Hospital, Tokyo, Japan, between 1999 and 2011 [17]. Follow-up was conducted between 2000 and 2017 (median, 78.05 months; range, 7.2–188.4 months). Tumor histological type was classified according to the International Union Against Cancer (UICC) TNM Classification of Malignant Tumors, 8th edition. Histological grading was performed as described previously [17].

The study was performed with the understanding and written consent of each patient. The study was approved by the Internal Review Board of the NCC (Approval No.: NCC, 2010-077) and carried out in accordance with the principles of the Declaration of Helsinki.

### 2.2. Antibodies

The following primary antibodies were used: anti-pan-cytokeratin (AE1/AE3) mouse monoclonal antibody (Dako, Santa Clara, CA, USA), anti-RIOK2 rabbit polyclonal antibody (HPA005681, Sigma Aldrich, St. Louis, MO, USA), anti–β-actin mouse monoclonal antibody (AC-15, Abcam, Cambridge, UK), and anti-S6 Ribosomal Protein rabbit monoclonal antibody (5G10, Cell Signaling Technology, Beverly, MA, USA), and Alexa Fluor® 488 Anti-Calreticulin rabbit monoclonal antibody (EPR3924, Abcam).

### 2.3. Tissue Microarrays and Fluorescence Immunohistochemistry

Tissue microarrays (TMAs) were constructed from FFPE pathological blocks. A tissue core 2 mm in diameter was removed from each tumor block and placed into recipient paraffin blocks using a tissue microarrayer [18].

Anti-rabbit IgG Alexa 594 (Invitrogen, Carlsbad, CA, USA) and anti-mouse IgG Alexa 488 (Invitrogen) antibodies were used as secondary antibodies. The imaging data analysis was performed as described previously [16]. Briefly, the sections were stained with pan-cytokeratin (green) to label cancer cells. We then changed the target spot image to binary image data by using only the green color component. The expression levels of RIOK2 (red) in cancer regions were quantified as the intensity of the red fluorescence using the

AQVIP system that was developed by our group. Cut-off values were calculated using X-tile algorithms based on the expression levels of RIOK2 [16,19].

### 2.4. Cell Culture

KOSC-2 cl3-43 (KOSC-2), HSC-2, HSC-3, HSC-4, and SCC-4 human oral cancer cell lines were obtained from the Japanese Collection of Research Bioresources Cell Bank (Osaka, Japan). KOSC-2 cells were grown in RPMI medium (Thermo Fisher Scientific, Waltham, MA, USA). HSC-2, HSC-3, and HSC-4 cells were grown in MEM (Thermo Fisher Scientific). SCC-4 cells were grown in Dulbecco's Modified Eagle Medium (Thermo Fisher Scientific). All media were supplemented with 10% fetal bovine serum (Thermo Fisher Scientific).

### 2.5. siRNA Transfection

Lipofectamine 2000 (Invitrogen) was used for siRNA transfection according to the manufacturer's instructions. Briefly, 50 μmol siRNA was diluted in 250 μL Opti-MEM I Reduced Serum Medium (Thermo Fisher Scientific), and 5 μL Lipofectamine 2000 was diluted in 250 μL Opti-MEM I Reduced Serum Medium. The diluted Lipofectamine 2000 was mixed with the diluted siRNA, and then the Lipofectamine/siRNA complex was applied to cell lines. These cells were seeded into 6-well plates ($5.0 \times 10^5$ cells per well) before incubation for 48 h. We purchased siRNA specific to RIOK2, constructed by QIAGEN (QIAGEN, Venlo, The Netherlands): RIOK2-siRNA_1 (5′-ATGAAACGTTTCAGCTACGAA-3′) and RIOK2-siRNA_2 (5′-TAGGAAGAACCTCGTTTCGAA-3′). As negative control siRNA, we used AllStars Negative Control siRNA (QIAGEN, Venlo, The Netherlands). Cells were harvested 48 h after transfection and used in further analysis.

### 2.6. Western Blotting

Reduced and boiled samples (typically 5–20 μg total protein per assay) were subjected to sodium dodecyl sulfate-polyacrylamide gel electrophoresis (SDS-PAGE) and transferred onto polyvinylidene difluoride (PVDF) membranes (Merck Millipore, Darmstadt, Germany). Antibody hybridization and immunoblotting detection were performed using previously described methods [20]. The densitometry intensity of protein bands was calculated using Image J software (version 1.53 k). The densitometry reading of each band is shown in Supplementary Figures S2 and S3.

### 2.7. ATP-Based Cell Growth Assay

HSC-2 and KOSC-2 cell lines were transfected with siRNAs for RIOK2 knockdown. After 48 h of incubation, cells were seeded onto 96-well plates ($1 \times 10^3$ cells per well) then incubated for 24–96 h. Cell growth was evaluated using the CellTiter-Glo 2.0 kit (Promega, Madison, WI, USA) according to the manufacturer's instructions. Luminescence was measured using the GloMax® Discover System (Promega). Cell growth was calculated relative to the corresponding luminescent signal intensity at 0 h. At 96 h after the start of measurement, cells were observed under a phase-contrast microscope (Nikon Instruments Inc., Tokyo, Japan).

### 2.8. Clonogenic Assay

To assess the colony formation capacity of oral cancer cell lines, HSC-2 and KOSC-2 cell lines were transfected with siRNAs for RIOK2 knockdown. After 48 h, HSC-2 and KOSC-2 cells (3000 cells and 5000 cells, respectively) were seeded into 6-well plates and incubated for 6 days with media replaced every 3 days. Cells were fixed with 4% formaldehyde for 10 min, and then stained with 2% crystal violet for 30 min. The stained cells were then rinsed with deionized water and air dried.

### 2.9. Protein Synthesis Assay

Measurement of protein synthesis was performed using a protein synthesis assay kit (Cayman, Ann Arbor, MI, USA) according to the manufacturer's instructions. Briefly,

oral cancer cell lines were transfected with RIOK2 siRNAs and negative control siRNAs. O-Propargyl-Puro (OPP) labeling was performed 6 days after transfection. The experiment was performed in triplicate starting from three independently transduced wells. Cells were labeled with OPP after transfection with siRNA against RIOK2. The translation inhibitor cycloheximide (CHX) (1 μg/mL) was used as a control for the inhibition of protein synthesis. The cell growth assay was performed simultaneously to normalize total cell number between the samples.

### 2.10. Statistical Analysis

EZR (Saitama Medical Center, Jichi Medical University, http://www.jichi.ac.jp/saitama-sct/SaitamaHP.files/statmedEN.html, accessed on 1 November 2022), which is a graphical user interface for R (The R Foundation for Statistical Computing, Vienna, Austria, version 2.13.0) was used for the data analysis [21]. The EZR interface is a modified version of R commander designed to add statistical functions that are frequently used in biostatistics. Associations between different categorical variables were assessed by Fisher's exact test. Overall survival (OS) was measured as the period from primary surgery to the date of death or last follow-up. Kaplan–Meier curves were used to estimate survival time and the log-rank test was used to analyze differences in survival. Cox regression models were used to calculate the hazard ratio (HR) and the 95% confidence interval (95%CI) between groups. Welch's $t$ test was used to analyze differences in the means of metric variables. A $p$-value of less than 0.05 was considered statistically significant.

## 3. Results

### 3.1. Prognostic Impact of RIOK2 Protein Expression in TSCC Patients

To investigate the relationship between RIOK2 expression and prognosis in patients with TSCC, we first stained TMAs for anti-RIOK2 and pan-cytokeratin antibodies (Figure 1A). We then quantified RIOK2 expression in the pan-cytokeratin-positive cancer regions using AQVIP as described previously [16]. We calculated the cut-off value of RIOK2 expression using X-tile algorithms [16,19] to determine RIOK2-positive ($n = 14$) and -negative ($n = 26$) cases and tested their correlation with the clinicopathological parameters (Figure 1B). RIOK2-positive cases were positively correlated with poor/moderate histological differentiation ($p = 0.0428$) (Table 1). Kaplan–Meier analysis showed that RIOK2 expression in cancer cells was significantly associated with poor prognosis in TSCC patients (log-rank $p = 0.013$) (Figure 1C). We also observed a trend toward worse disease-free survival (DFS) in RIOK2-positive cases (Supplementary Figure S1, $p = 0.101$, log-rank test). No statistically significant association was observed between RIOK2 and age, sex, clinical stage, mode of invasion, perineural invasion, or lymphovascular invasion (Table 1).

Univariate regression analysis revealed that RIOK2 expression and lymphovascular invasion were significantly associated with prognosis (HR, 3.752; 95%CI, 1.225–11.49; $p = 0.02063$ and HR, 3.130; 95%CI, 1.017–9.632; $p = 0.04660$, respectively) (Table 2). Multivariate analysis indicated that RIOK2 expression was an independent prognostic factor in TSCC patients (HR, 3.539; 95%CI, 1.149–10.91; $p = 0.02772$) (Table 2).

### 3.2. Effects of RIOK2 Expression on Oral Cancer Cell Growth

We next sought to understand the function of RIOK2 in TSCC progression. To examine the effect of RIOK2 expression on cell growth, we chose HSC-2 and KOSC-2 oral cancer cell lines that possessed the highest endogenous RIOK2 expression (Figure 2A). Treatment with two different siRNA sequences successfully reduced protein expression of RIOK2 in the KOSC-2 and HSC-2 cell lines (Figure 2B). In addition, treatment with RIOK2 siRNAs resulted in a statistically significant decrease in cell growth in both cell lines compared to treatment with the negative control siRNA (Figure 2C). To exclude the possibility that RIOK2 ATPase activity could have affected the result of the ATP-based cell growth assay [9], we also confirmed the cell number of each condition microscopically (Figure 2D). In addition, we used a clonogenic assay to assess the effect of siRIOK2 on the colony formation

capacity of these oral cancer cell lines [22]. RIOK2 knockdown strongly reduced the colony formation capacity of oral cancer cell lines (Figure 2E), suggesting that RIOK2 expression is essential for cell growth in oral cancer cell lines.

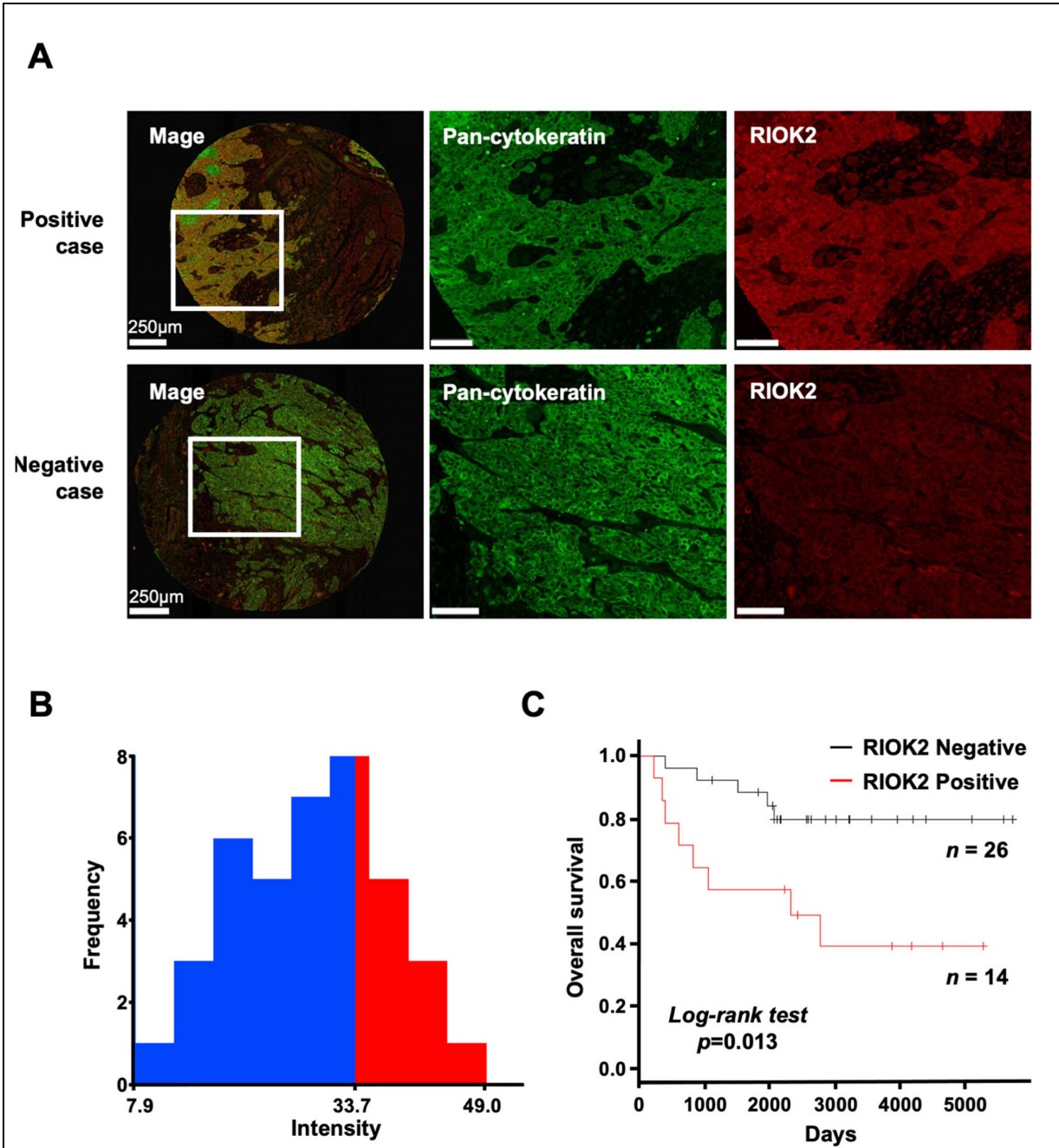

**Figure 1.** Relationship between RIOK2 expression and patient prognosis using 40 TSCC tissue samples. (**A**) Immunofluorescence staining with anti-pan-cytokeratin (green) antibody and anti-RIOK2 (red). Representative images of a positive case are shown in the upper panels and those of a negative case are shown in the lower panels. The scale bars indicate 250 μm in the whole images (left panel) and 100 μm in the middle and right panels. (**B**) Histogram shows the intensity of anti-RIOK2 fluorescence in individual patients, and the cut-off values for defining high-intensity (red) and low-intensity (blue) groups are shown. Differences were analyzed by log-rank test. (**C**) OS curves for the RIOK2-positive (red line, $n = 14$) and RIOK2-negative (black line, $n = 26$) subgroups ($p = 0.013$).

**Table 1.** Association of RIOK2 with clinicopathologic characteristics in stage I/II TSCC.

| | | | RIOK2 IHC | | |
|---|---|---|---|---|---|
| | | Number of Cases (%) | Negative | Positive | *p*-Value [a] |
| Total | | 40 (100) | 26 | 14 | |
| Age (years) | | | | | |
| | <62 | 19 (47.5) | 15 | 4 | 0.1050 |
| | ≥62 | 21 (52.5) | 11 | 10 | |
| Sex | | | | | |
| | Male | 15 (37.5) | 7 | 8 | 0.08940 |
| | Female | 25 (62.5) | 19 | 6 | |
| Stage [b] | | | | | |
| | I | 21 (52.5) | 14 | 7 | 0.1860 |
| | II | 19 (47.5) | 12 | 7 | |
| Histologic differentiation | | | | | |
| | Well | 26 (65) | 20 | 6 | **0.04280 *** |
| | Poor/Moderate | 14 (35) | 6 | 8 | |
| Mode of invasion [c] | | | | | |
| | 1,2 | 23 (57.5) | 17 | 6 | 0.1980 |
| | 3,4 | 17 (42.5) | 9 | 8 | |
| Perineural invasion | | | | | |
| | Negative | 39 (97.5) | 25 | 14 | 1.000 |
| | Positive | 1 (2.5) | 1 | 0 | |
| Lymphovascular invasion | | | | | |
| | Negative | 31 (77.5) | 22 | 9 | 0.2340 |
| | Positive | 9 (22.5) | 4 | 5 | |

Correlation between RIOK2 protein expression and clinicopathologic characteristics in stage I/II tongue squamous cell carcinoma. [a] Fisher's exact test, * *p* < 0.05. Statistically significant differences are highlighted in bold. [b] According to the Union for International Cancer Control (UICC) TNM Classification of Malignant Tumors, 8th edition. [c] Anneroth's histological grading system.

**Table 2.** Hazard ratios for deaths due to stage I/II tongue cancer; Cox regression model.

| Covariate | | Univariate Analysis (*n* = 40) | | | Multivariate Analysis (*n* = 40) | | |
|---|---|---|---|---|---|---|---|
| | | HR | 95%CI | *p*-Value | HR | 95%CI | *p*-Value |
| Age (year) | | | | | | | |
| | <62 | | Reference | | | | |
| | ≥62 | 3.514 | 0.9648–12.80 | 0.05671 | | | |
| Sex | | | | | | | |
| | Male | | Reference | | | | |
| | Female | 0.9959 | 0.3248–3.054 | 0.9943 | | | |
| Stage [a] | | | | | | | |
| | I | | Reference | | | | |
| | II | 1.445 | 0.4850–4.306 | 0.5086 | | | |
| Differentiation | | | | | | | |
| | Well | | Reference | | | | |
| | Poor/ Moderate | 2.527 | 0.8459–7.547 | 0.09686 | | | |
| Mode of invasion [b] | | | | | | | |
| | 1,2 | | Reference | | | | |
| | 3,4 | 2.620 | 0.8539–8.036 | 0.09686 | | | |
| Lymphovascular invasion | | | | | | | |
| | Negative | | Reference | | | Reference | |
| | Positive | 3.130 | 1.017–9.632 | **0.04660 *** | 2.871 | 0.9218–8.941 | 0.06884 |
| RIOK2 IHC | | | | | | | |
| | Negative | | Reference | | | Reference | |
| | Positive | 3.752 | 1.225–11.49 | **0.02063 *** | 3.539 | 1.149–10.91 | **0.02772 *** |

HR, hazard ratio; CI, confidence interval; RIOK2, Rio-kinase 2. * *p* < 0.05. Statistically significant differences are highlighted in bold. [a] According to the Union for International Cancer Control (UICC) TNM Classification of Malignant Tumors, 8th edition. [b] Anneroth's histological grading system.

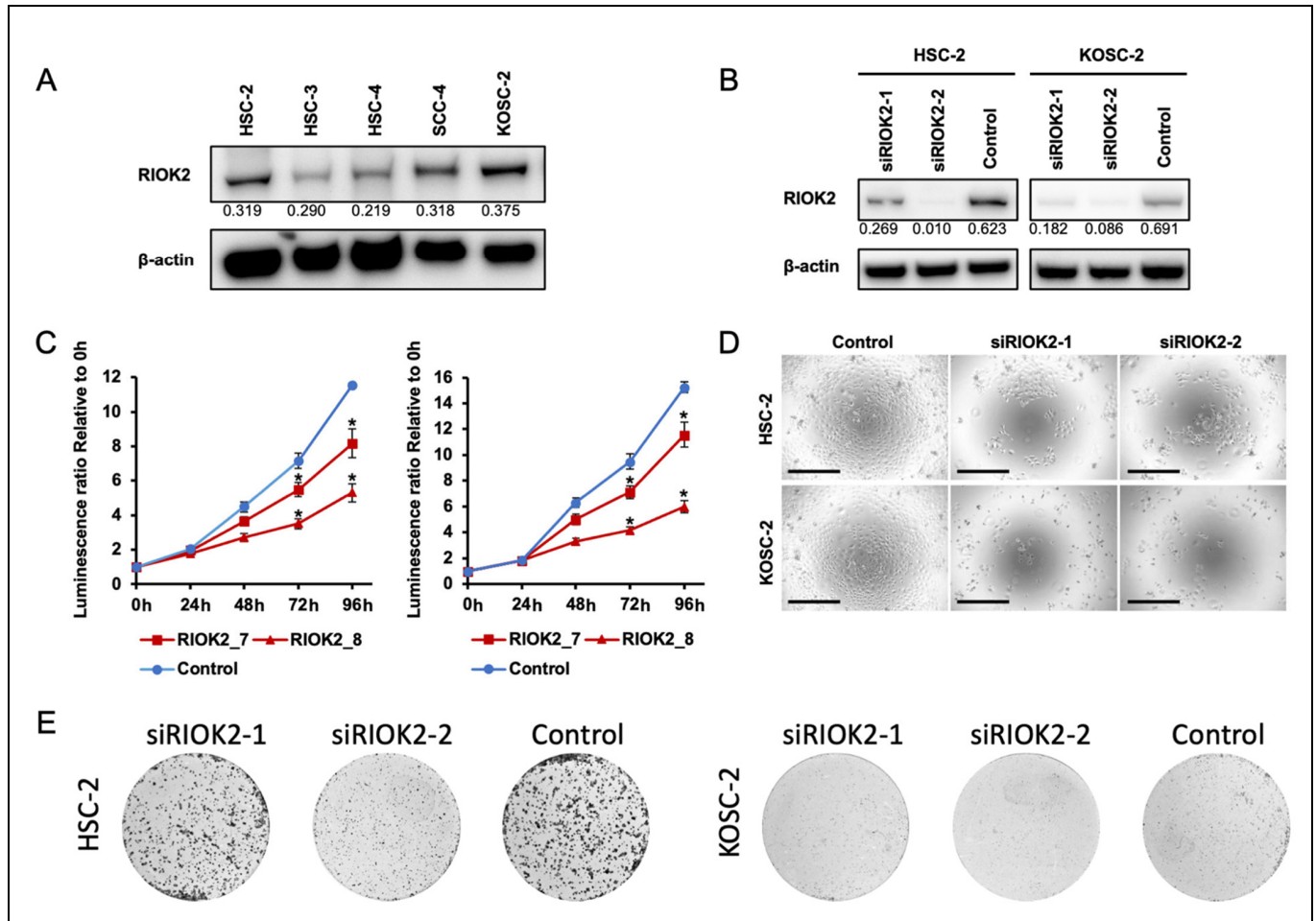

**Figure 2.** RIOK2 expression is involved in oral cancer cell growth. (**A**) Western blot analysis of RIOK2 expression was performed in two oral cancer cell lines transfected with siRIOK2 (siRIOK2-1, siRIOK2-2) and siRNA Control (control). Cell lysates were prepared 48 h (for KOSC-2) and 96 h (for HSC-2) after transfection. β-actin expression was blotted as a loading control. The densitometry intensity/reading ratio (RIOK2/β-actin) is indicated under the band image. (**B**) Western blot analysis of RIOK2 expression in two oral cancer cell lines transfected with siRIOK2 (siRIOK2-1, siRIOK2-2) and siRNA Control (control). β-actin expression was blotted as a loading control. The densitometry intensity/reading ratio (RIOK2/β-actin) is indicated under the band image. (**C**) Cell growth was evaluated in siRIOK2-1, siRIOK2-2, and Control OSCC cell lines (KOSC2, HSC2) at the indicated time points, and (**D**) representative photographs of transfected cells after 96 h of culture are shown. The scale bars indicate 500 μm. Optical densities at 490 nm were measured using plate reader. Welch's *t*-test was used to assess statistical significance (* $p < 0.05$). Technical replicates, $n = 6$; biological replicates, $n = 3$. (**E**) Colony formation capacity was assessed using HSC-2 and KOSC-2 transfected with siRIOK2 and siRNA Control (control). Representative images are shown here. Technical replicates, $n = 3$, from two independent experiments.

### 3.3. RIOK2 Regulated Protein Synthesis in Oral Cancer Cells

We found that siRNA targeting RIOK2 reduced S6 protein expression in HSC-2 and KOSC-2 (Figure 3A). S6 protein is one of the compartments of the 40S ribosomal subunit in eukaryote cells [23], which suggests that RIOK2 knockdown affects ribosomal biogenesis in oral cancer cell lines. Therefore, we hypothesized that RIOK2 knockdown leads to decreased cell growth by inhibiting ribosome maturation and protein synthesis. To examine the effect of RIOK2 expression on protein synthesis, we analyzed the global translation capacity using puromycin analogue OPP [24]. OPP bears a terminal alkyne group that allows the copper(I)-catalyzed azide-alkyne cycloaddition (CuAAC) reaction [24]. OPPs incorporated into the C

terminus of translating polypeptide chains are subsequently detected via a CuAAC reaction with 5-carboxyfluorescein (5 FAM)-azide. We added OPP to the oral cell lines 6 days after transfection of siRNAs, then quantified the global translation capacity of each cell line using the fluorescence intensity of 5 FAM-azide. As expected, RIOK2 knockdown led to a significant decrease in protein synthesis of oral cancer cell lines (Figure 3B).

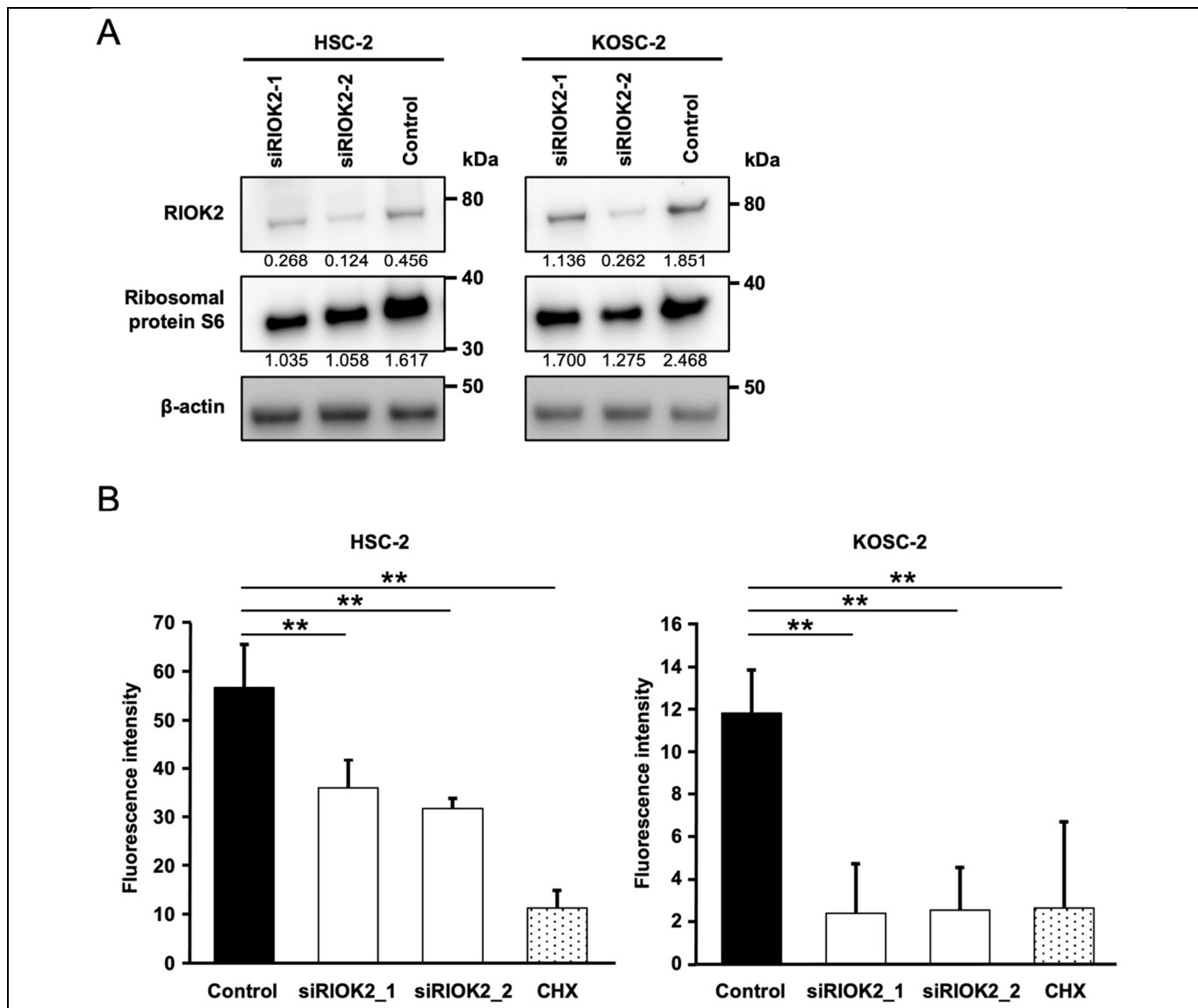

**Figure 3.** RIOK2 expression affects protein synthesis of the OSCC cell line. (**A**) Western blot analysis of RIOK2 and S6 ribosomal protein in OSCC cell lines transfected with RIOK2 siRNAs. Cell lysates were prepared 6 days after transfection. In this analysis, 5 μg samples of total protein were used. The densitometry intensity/reading ratio (RIOK2/β-actin and S6/β-actin) is indicated below the band image. (**B**) O-Propargyl-puromycin labeling of HSC-2 and KOSC-2 cells transfected after 6 days. Fluorescence intensity of 5 FAM-azide was normalized by CellTiter-Glo assay data. Welch's *t*-test was used to assess statistical significance (** $p < 0.01$). The translation inhibitor cycloheximide (1 μg/mL) was used as a positive control. Technical replicates, $n = 8$.

## 4. Discussion

Selection of the treatment strategies for TSCC patients remains a challenge because both cancer resection and preservation of facial function must be considered. Thus, more accurate biomarkers for predicting patient outcomes are required in order to make decisions on effective therapeutic strategies for TSCC. In the present study, immunohistochemical

analysis indicated that RIOK2 is a potential biomarker for predicting prognosis in patients with TSCC. The finding that RIOK2 expression was strongly associated with ribosomal function in TSCC cells indicates that it is suitable for use as a precision biomarker for ribosome-targeted therapy. It is likely that not all TSCCs require aberrant ribosome function because our data also showed the presence of TSCC cells with low RIOK2 expression. However, RNA polymerase I inhibitor CX5461 effectively induces apoptosis in oral cancer cell lines [25]. Although further examination should be performed to evaluate the therapeutic effect of ribosome targeting on TSCC progression, the present findings demonstrate the potential of ribosome targeting to improve diagnostic and therapeutic strategies for TSCC.

There is growing evidence that RIOK2 is a potential target for cancer treatment. A previous study reported that RIOK2 formed complexes with RIOK1 and mTOR and then enhanced the Akt-signaling pathway in glioblastoma [26]. Inhibition of RIOK1 or RIOK2 expression diminished Akt signaling and induced cell-cycle arrest, apoptosis, and chemosensitivity in glioblastoma cells via the p53-dependent pathway [26]. The overexpression of RIOK2 in non-small cell lung cancer (NSCLC) was associated with poor clinical outcomes, clinical stage, differentiation, and lymph node metastasis [12]. Some tumor-suppressive miRNAs can also target RIOK2 expression in lung cancer and glioma [13,14]. Consistent with these previous studies, we showed that RIOK2 expression was significantly correlated with a worse outcome in TSCC patients and that a decrease in RIOK2 expression affected cell growth in oral cancer cell lines. Several recent studies have described the effect of pharmacological inhibition of RIOK2 expression in prostate cancer, glioblastoma, and acute myeloid leukemia (AML) [15,27,28]. Therefore, our findings and those of previous studies collectively imply that use of RIOK2 inhibitory agents are an attractive strategy for TSCC therapy.

Several reports have indicated that inhibition of RIOK2 expression induced ribosomal stress in prostate cancer [28] and ribosome dysfunction in AML [15]. The present study showed the effect of RIOK2 expression on S6 ribosomal protein expression and protein synthesis in oral cancer cell lines, suggesting that RIOK2 inhibition can induce ribosomal stress in oral cancer cell lines. As the regulator of cellular translation, S6 protein and the eukaryotic initiation factor 4E (eIF4E) are well-studied [29]. The precise mechanisms by which RIOK2 expression mediates cell growth and protein synthesis in oral cancer remain unknown. However, it is plausible that RIOK2 contributes to ribosomal function via mediation of ribosomal protein expression. Further examination to completely clarify the molecular mechanisms and function of RIOK2 expression in oral cancer is therefore needed.

A limitation of our study is the small number of TSCC cases. First, we were unable to show a statistically significant correlation between RIOK2 expression and DFS ($p = 0.101$ log-rank test, Supplementary Figure S1). Second, the present univariate analyses of 40 TSCC cases did not identify the mode of invasion as an independent prognostic factor (HR, 2.62; 95%CI, 0.8539–8.036; $p = 0.09686$) (Table 1). In general, the mode of invasion is an adverse prognostic factor of TSCC. RIOK2 expression could potentially indicate some biological aspect of TSCC that is independent of the mode of invasion. However, further statistical analysis in a large cohort of patients is required to validate RIOK2 as a robust prognostic biomarker.

In conclusion, RIOK2 expression is associated with poor prognosis in TSCC patients. Our data also suggest that RIOK2 expression contributes to oral cancer cell growth, ribosome biogenesis, and protein synthesis. We believe that our findings will lead to an understanding of the molecular basis of TSCC and to improvements in cancer treatment and diagnosis.

**Supplementary Materials:** The following supporting information can be downloaded at: https://www.mdpi.com/article/10.3390/curroncol30010031/s1, Figure S1: Relationship between RIOK2 expression and disease-free survival (DFS), Figure S2: Uncropped full-length images of western blot analysis and the densitometry intensity/reading ratio in Figure 2, Figure S3: Uncropped full-length images of western blot analysis and the densitometry intensity/reading ratio in Figure 3.

**Author Contributions:** Conceptualization, Y.M., Y.N., T.S., M.T. and K.H.; methodology, Y.M., Y.N., N.M, T.M., Y.W. and K.H.; formal analysis, Y.M., Y.N., N.M. and Y.W.; resources, T.M. and S.Y.; data curation, Y.M., Y.N., N.M. and Y.W.; writing—original draft preparation, Y.M., Y.N. and K.H.; writing—review & editing, Y.M., Y.N. and K.H.; supervision, Y.N., T.S., M.T. and K.H.; project administration, Y.N. and K.H.; funding acquisition, K.H. All authors have read and agreed to the published version of the manuscript.

**Funding:** This work was funded and supported by a Grants-in-Aid for Scientific Research (B) (16H05547) from the Ministry of Education, Culture, Sports, Science and Technology (METX) of Japan (K.H.), and by Therapeutic Evolution (P-CREATE) (21cm0106403h0006) from the Japan Agency for Medical Research and Development (K.H.).

**Institutional Review Board Statement:** The study was performed with the understanding and written consent of each patient. The study was approved by the Internal Review Board of the National Cancer Center (approval number: NCC, 2010-077) and carried out in accordance with the principles of the Declaration of Helsinki.

**Informed Consent Statement:** The requirement for informed consent was waived by the committee because of the retrospective nature of the study.

**Data Availability Statement:** The data presented in this study are available on request from the corresponding author upon reasonable request.

**Acknowledgments:** We thank K. Yoshida for his advice on the biological aspects of the study, and T. Toyoda and R. Tokita for their assistance with the experiments. We extend special thanks to K. Takeuchi for secretarial assistance.

**Conflicts of Interest:** The authors declare no conflict of interest.

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
