# Peer review of "RIOK2 Contributes to Cell Growth and Protein Synthesis in Human Oral Squamous Cell Carcinoma"

_curroncol, doi:10.3390/curroncol30010031_

Round 1
Reviewer 1 Report
The study conducted by Yusuke Matsuzaki entitled “RIOK2 contributes to cell growth and protein synthesis in human oral squamous cell carcinoma” reported role of RIOK2 in human oral squamous cell carcinoma. Author reported that expression of RIOK2 as a key player in pre-40S ribosomal complex maturation and it is associated with poor overall survival in patients with TSCC
The study has been designed properly and most of the results and their conclusions are very convincing and sound. Still, some experiments would help the readers grab the story better. As per my consideration, it needs major revision before final acceptance.
Comments
1. As author mentioned RIOK2 contributes to cell growth and protein synthesis in human oral squamous cell carcinoma but they have not validated this experimentally, I would recommend testing the effect of RIOK2 overexpression and knockdown on global translation by using puromycin incorporation assay as mentioned in this article (https://pubmed.ncbi.nlm.nih.gov/29618122/).
2. Fig. 3A the knockdown of RIOK2 for siRIOK2-1 in KOSC-2 cell line don’t look good but the viability assay showed significant reduction in the cell viability. Please explain??
3. Author performed cell viability assay which suggests RIOK2 depletion reduces the cell viability. I would suggest to test the effect of RIOK2 depletion on clonogenic potential of the cell as mentioned in (https://pubmed.ncbi.nlm.nih.gov/30176153/).
Reviewer 2 Report
It is known that the ribosome is responsible for protein synthesis, which maintains cellular homeostasis and is necessary for the rapid cell division of cancer cells. However, it remains unclear how mediators of ribosome biogenesis are involved in the malignant behavior of tongue squamous cell carcinoma (TSCC). The authors found that expression of RIOK2, a key player in the maturation steps of the ribosome complex up to 40S, was significantly associated with lower overall survival in TSCC patients.
1. All participants in the study underwent only a glossectomy. In the case of lymphovascular invasion, the patients did not receive radiation and chemotherapy? The type of treatment may also affect survival rates and should be considered.
2. No information on comorbidities, smoking status, etc.
3. Which patients relapsed? Did all patients die from the underlying disease? The authors provide only data on overall survival, but not on disease-free survival.
Round 2
Reviewer 1 Report
Author addressed all raised queries so i recommend to accept the manuscript in current form.
Author Response
Thank you very much.
Reviewer 2 Report
The authors answered questions from reviewers and made significant changes to the text of the manuscript. I believe that in its present form the manuscript can be recommended for publication.
Author Response
Thank you very much.